# Visualization and validation of twin nucleation and early-stage growth in magnesium

Lin Jiang[1,2], Mingyu Gong[3], Jian Wang[4], Zhiliang Pan[1], Xin Wang[1], Dalong Zhang[1], Y. Morris Wang [5], Jim Ciston [6], Andrew M. Minor [6,7], Mingjie Xu[1], Xiaoqing Pan[1,8], Timothy J. Rupert[1], Subhash Mahajan[9], Enrique J. Lavernia [10], Irene J. Beyerlein [11] & Julie M. Schoenung [1✉]

The abrupt occurrence of twinning when Mg is deformed leads to a highly anisotropic response, making it too unreliable for structural use and too unpredictable for observation. Here, we describe an in-situ transmission electron microscopy experiment on Mg crystals with strategically designed geometries for visualization of a long-proposed but unverified twinning mechanism. Combining with atomistic simulations and topological analysis, we conclude that twin nucleation occurs through a pure-shuffle mechanism that requires prismatic-basal transformations. Also, we verified a crystal geometry dependent twin growth mechanism, that is the early-stage growth associated with instability of plasticity flow, which can be dominated either by slower movement of prismatic-basal boundary steps, or by faster glide-shuffle along the twinning plane. The fundamental understanding of twinning provides a pathway to understand deformation from a scientific standpoint and the microstructure design principles to engineer metals with enhanced behavior from a technological standpoint.

[1] Department of Materials Science and Engineering, University of California, Irvine, CA 92697, USA. [2] Materials & Structural Analysis Division, Thermo Fisher Scientific, Hillsboro, OR 97124, USA. [3] State Key Lab of Metal Matrix Composites, Shanghai Jiao Tong University, Shanghai 200240, China. [4] Department of Mechanical & Materials Engineering, University of Nebraska-Lincoln, Lincoln, NE 68588, USA. [5] Department of Materials Science and Engineering, University of California, Los Angeles, Los Angeles, CA 90095, USA. [6] Molecular Foundry, Lawrence Berkeley National Laboratory, Berkeley, CA 94701, USA. [7] Department of Materials Science and Engineering, University of California, Berkeley, CA 94720, USA. [8] Department of Physics and Astronomy, University of California, Irvine, CA 92697, USA. [9] Department of Materials Science and Engineering, University of California, Davis, CA 95616, USA. [10] National Academy of Engineering, Irvine, CA 92617, USA. [11] Department of Mechanical Engineering and Materials, University of California, Santa Barbara, CA 93101, USA. ✉email: schoenuj@uci.edu

Magnesium (Mg), being 4.6 times lighter than steel, has been recognized for more than a century as a potentially energy-saving, lightweight structural material for load-bearing applications[1–6]. Twinning is a common deformation mechanism in Mg, because Mg does not have a sufficient number of easy slip systems[2,4–6]. However, deformation twinning results in strong plastic anisotropy and low ductility in Mg due to the localized, unidirectional shear deformation[1–6]. Numerous studies for over half a century have intensely examined the deformation conditions, microstructural effects, and alloying effects that tend to promote or suppress the occurrence of twinning[1,7–10]. Yet, the mechanisms responsible for the nucleation and early-stage growth of deformation twins remain to be clarified.

A major controversy concerning the description of $\{\bar{1}012\}$ $<10\bar{1}1>$ twinning (the most common type in hexagonal close packed, HCP, metals) is whether it follows the conventional shear-shuffle nucleation mechanism (such as the pole mechanism involving gliding of twin dislocations on the twinning plane) or the proposed pure-shuffle nucleation mechanism (classical phase transformation mechanism)[6,11–13]. However, manipulation or isolation of a twin nucleation event for direct visualization has proven technically challenging[14,15]. Deformation twinning is a localized, ultrafast (might be supersonic[16,17]) and stochastic event that, in principle, relies on exceptional points in the microstructure, where intense local stresses and certain types and density of dislocations coincide in time and space[12,13]. Although twin formation under the stress concentration introduced by nanoindentation has been predicted, simulated, or even intuitively expected[6,12,13], the sequence of events leading to the formation of the visible twins at an early-stage in Mg have not been tracked. Consequently, for several decades, twin nucleation in Mg has only been addressed by theory and atomic scale simulation, requiring presuppositions on the initiating defect and how twins first expand[18–26].

In this study, we use a combination of nanomechanical deformation, in-situ transmission electron microscopy, and atomic-scale simulation to isolate and identify the nucleation and early-stage growth processes of $\{10\bar{1}2\}$ deformation twins in Mg. We strategically designed truncated wedge-shaped pillars (TWPs) from single-crystal Mg to generate a steep stress field in the crystal under compression that supports twin nucleation but not rapid pillar-wide twin propagation and growth. We capitalized on the fact that twins prefer to nucleate in regions of high stress concentration, using our geometrical TWP design to confine twin nucleation in a nanosized region for visualization. We studied geometry effects contributing to the formation of a smaller twin nucleus within the pillar top. With a decrease in the width of the pillar top, the probability of suppressing initial growth of the twin nucleus increases, allowing for the possibility that we can characterize a twin nucleus and its early-stage growth within a submicron-sized pillar top. For the crystals with a regular rectangular geometry, only an "adult" twin was observed, and its early-stage growth likely occurred via fast shear-shuffling along the twinning plane. For the wedge-shaped crystals with larger tops (450 nm), we observed two co-zone twins emitting from each corner and quickly shearing along the twinning plane. For the samples with smaller tops (250 nm, 100 nm), we can track the twin tip movement and uniquely reveal that nucleation and growth of a stable embryo unfolds in multiple stages. First, the twin embryo is seen to originate at the contacted compression platen/pillar surface with its tip parallel to the basal plane. Then it was found to gradually expand along the basal plane, instead of shearing along the twinning plane. From atomic simulation and atomic resolution image analysis, the twin/matrix boundaries at all stages in the nucleation and early-stage growth process, are fully faceted and migrate by atomic shuffling. In remarkable contradiction to conventional belief, these early stages do not involve creation of the $\{10\bar{1}2\}$ coherent twin boundary, a twinning dislocation, or even basic shear displacements along the conventional twinning shear direction.

## Results

### Stress differences enabled isolation and in-situ visualization of twin nucleation and growth

When the rectangular pillar without an engineered stress differences is compressed, the stress state generated in the pillar is nearly uniform (Fig. 1a). As the load is applied, twins are often seen to appear suddenly and to immediately propagate across the pillar. This is a common outcome and a likely indicator that the stress to form the twin is higher than that needed to propagate it. Once the stress in the pillar becomes sufficient to nucleate a twin, the stress is more than adequate to propagate the twin. The time scales over which the entire event occurred are too short to distinguish the nucleation process from the propagation process[1,27]. These observations are in agreement with recent in-situ TEM studies on deformation twins, using conventional pillars, in which the twin propagates extremely fast, so fast that the twinning process is completed within 0.04 s, and under these conditions only adult twin with a size over 100 nm can be observed[28]. Therefore, in regular pillars, the sequence of events leading to the formation of the visible twin were not tracked and, in all cases, the twins discussed in the references were already visible and stable with the "adult" shape as we found in the regular geometry pillars, and hence believed not to represent a twin nucleus.

To form a twin nucleus but stunt its subsequent propagation, a high stress needs to be localized within a small region. The high stress in a small region must be sufficient to form the twin, while the low stress in the surrounding region must be insufficient to expand, or grow, the twin. To this end, we designed pillars with a truncated wedge-shaped geometry, wherein the top width of the pillar is narrower than the main pillar width. Figure 1 shows three TWP geometries, with top widths of 400, 250, and 100 nm. In all, the inclination angle is 45 degrees and the main pillar width is 750 nm. By comparison, the width of the conventional, constant width pillar is 750 nm.

To estimate the stress field generated when these TWP geometries are deformed, we employed finite element analysis (FEA) and simulated pillar deformation under an applied axial compression, just as in the experimental configuration. Maps for the shear stress on the two $\{10\bar{1}2\}$ twinning planes and internal compressive stress component acting on the prismatic slip planes were analyzed, as these components can be used as driving forces for twin formation and twin boundary migration. Supporting evidence for the relationship between these stress components and the mechanisms of twin formation are presented in Supplementary Figs. 1–3 and in Supplementary Discussion.

The FEA calculations show that the stress is distributed homogeneously in a conventional pillar (Fig. 1a), but heterogeneously in the TWPs. The distribution is characterized by a high stress region localized at the top of the pillar, as shown in Fig. 1b–d and Supplementary Fig. 2. The characteristics depend on the top width of the truncated wedge, with the locations of stress concentration points gradually moving from the two corners for the largest truncated wedge top (400 nm) to the center for the smallest truncated wedge top (100 nm). Concomitantly, as the width of the top is made narrower, the high stress region becomes more confined to the region in the upper portion of the truncated wedge.

To quantify the stress variation further experimentally within the pillars, we utilize normalized experimental compressive stress at different strains, as shown in Fig. 1i. Clearly, the averaged stress

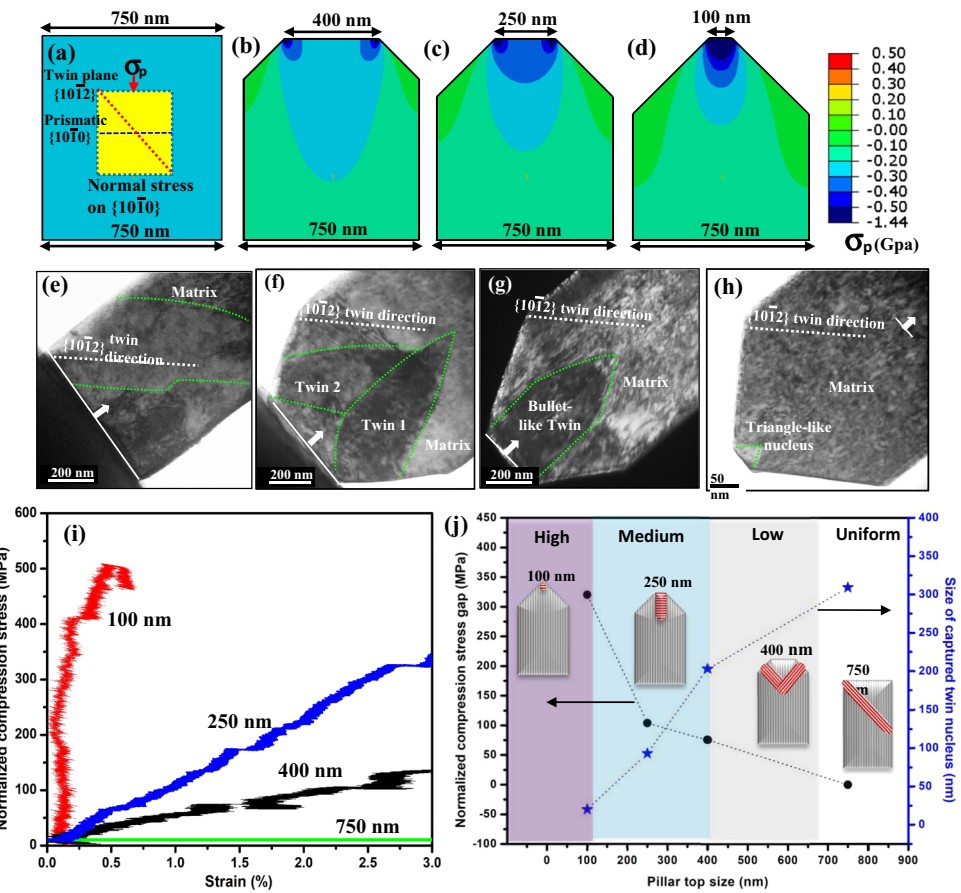

**Fig. 1 Twinning behavior and stress fields in single-crystal Mg pillars with various geometries. a–d** Normal stress on the prismatic planes ($\sigma_p$) and **e–h** twin morphologies with respect to the pillars: **e** a gigantic twin lamella within a conventional pillar; **f** two intersected co-zone twins within a truncated wedge-shaped pillar with 400 nm top; **g** one bullet-shaped twin within a truncated wedge-shaped pillar with 250 nm top; **h** one triangle-like twin nucleus within a truncated wedge-shaped pillar with 100 nm top. The arrows in **e–h** indicate the compression direction. **i** Normalized compression stress curves for the various pillars with different top sizes for the in-situ tests. The compression stress curves were normalized on the basis of the gap between the averaged stress at the truncated wedge pillar top (max. stress) and averaged stress at the pillar bottom (min. stress) at different strains and were calculated by using the max. stress curves subtracting the min. stress curves, as shown in Supplementary Fig. 3i–l. **j** Measured normalized compressive stress gap and critical sizes of the captured twin nucleuses at the twin nucleation points as a function of the pillar top width. The data in **i, j** were derived on the basis of Supplementary Fig. 3.

on the cross section of the pillar top is much higher than the averaged stress on the cross section of the pillar bottom, due to the truncated wedge shape of the pillars. At a given pillar bottom width, pillars with shorter top width have a larger stress difference between the top and bottom cross sections, indicating a higher stress concentration at the upper portion of the pillars. The compressive stress at a given strain is thus normalized by taking the maximum stress curves at the truncated wedge top and then subtracting the minimum stress curves at the truncated wedge bottom, as shown in Supplementary Fig. 3i–l. The results indicate that the magnitude of the normalized compressive stress is highest for the narrowest top. A soft normalized stress forms in the 400 nm top pillar, a mid-level stress localizes in the 250 nm top pillar, and a strong stress concentrates in the 100 nm top pillar, as shown in Fig. 1j. This agrees with the FEA-calculated stress distribution in different pillars, as shown in Fig. 1a.

To invoke and observe twin formation and early-stage growth, these pillar geometries were compressed in-situ in a TEM. For all pillar types, conventional and truncated wedge-shaped, twin formation is observed. (The in-situ testing is described in Supplementary Discussion).

For the rectangular pillars or pillars without a high stress localized in a small region, only a large twin nucleus was observed

to quickly propagate across the width of the pillar, as shown in Fig. 2a–d and Supplementary Fig. 5 (also see Supplementary Movie 1). Within the time increment of the test, the twin had already nucleated and grown. The smallest twin size captured at a total compressive strain of 4.85% (Supplementary Fig. 3e) is ~310 nm (using an equivalent circle diameter calculation), which is too large to be considered an embryo. The FEA suggests that the stress state generated in the pillar is nearly homogeneous, and the precise location of where this twin originated cannot be detected.

When the TWP has a 400 nm wide top, two stress concentration points are predicted to develop. Consistent with the peak stress points in the FEA calculation, in the compression experiment, we observe that two twins emanate from these corners, and expand until they intersect in the middle of the pillar, as shown in Supplementary Fig. 3b. Again, the nucleation and propagation steps occur too quickly to be distinguished and the sizes of the twins when they are first detected at a total compressive strain of 1.7% (Supplementary Fig. 3f) are ~200 nm.

In compression of the narrower truncated wedge top of 250 nm, a triangular twin is seen first and then, with increased displacement of the compression platen (Fig. 2l), it expands into a twin lamella, as shown in Fig. 2h–k and Supplementary Fig. 6

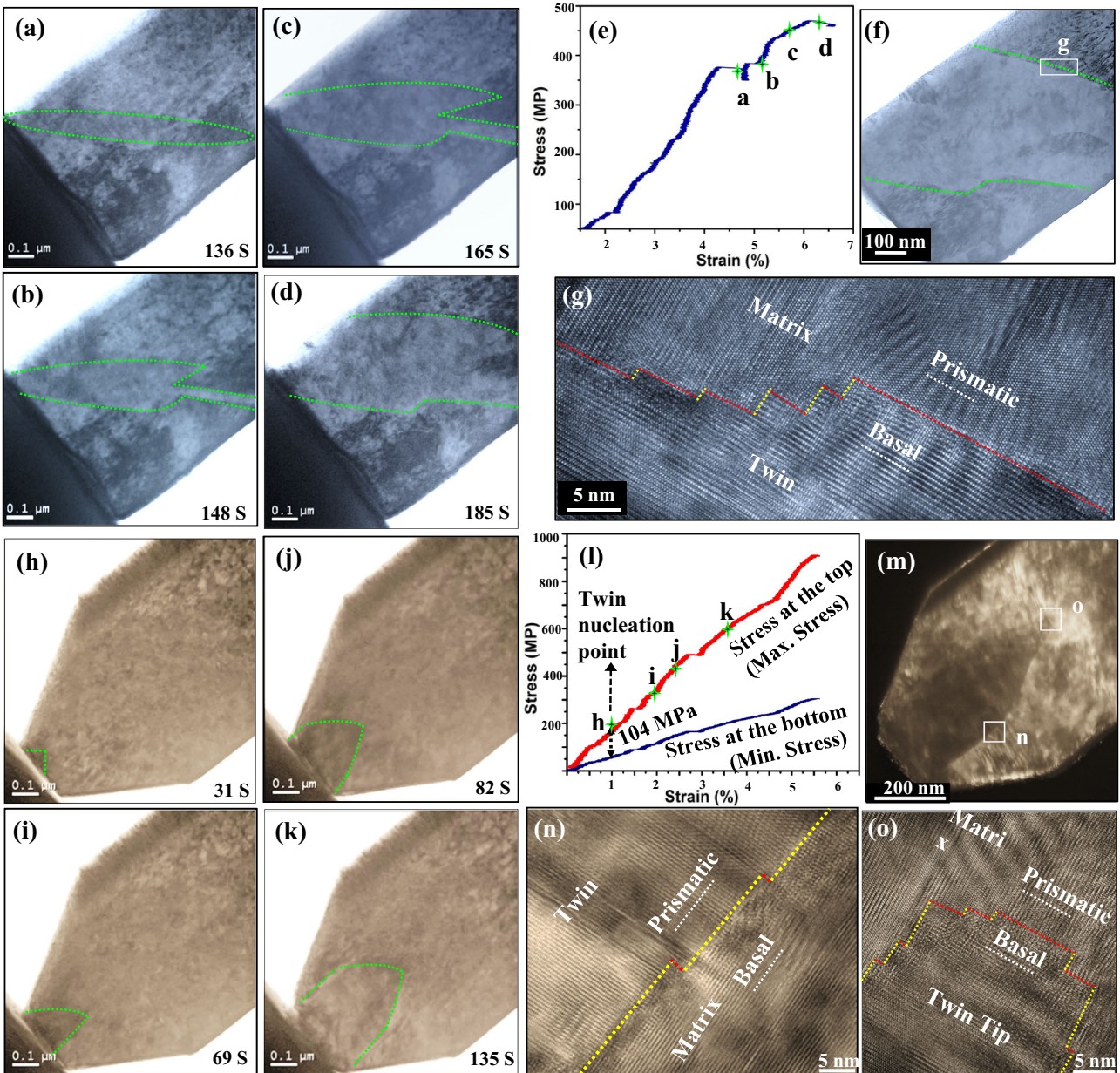

**Fig. 2 Twinning events in the Mg pillars with different stress gap magnitudes. a–g** twin growth in the conventional pillar with the width of 750 nm under a homogenous stress field: **a–d** evolution of twin with indentation time; **e** corresponding compression curves; **f** enlarged TEM image of the final twin; **g** HRTEM images of basal/prismatic terraced boundary. **h–o** Nucleation and initial growth of one twin nucleus in the truncated wedge-shaped pillar with the top width of 250 nm under a medium stress difference: **h–k** evolution of twin nucleus with indentation time; **l** corresponding compression curves; **m** enlarged TEM image of the final twin; HRTEM images of **n** basal/prismatic terraced boundary, and **o** twin tip. The yellow dashed lines in **g, n–o** indicate the basal-prismatic interfaces with the basal plane of the matrix parallel to the prismatic plane of the twin and the red lines in **g, n–o** indicate the prismatic-basal interfaces with the prismatic plane of the matrix parallel to the basal plane of the twin. The max. stress curve in **l** was calculated by the ratios of compression loading to the cross section of the pillar top. The min. stress curve in **l** was calculated by the ratios of compression loading to the cross section of the pillar bottom. The value of 104 MPa in **l** shows the stress gap at the twin nucleation point h.

(also see Supplementary Movie 2). Unlike the other two pillar designs, with the relatively higher stress gradient generated in this TWP, it is possible to distinguish the formation from subsequent early-stage growth. As seen in Fig. 2h, in the pillar with the 250 nm-wide top, the size of the triangular twin when it is first identified at a compressive strain of 0.7% (Supplementary Fig. 3k) is ~90 nm. Then, it expands its boundary along the basal plane, instead of shearing along the basal plane (Fig. 2i–k). TEM characterization (Fig. 2m–o) finds that the resulting twin after the compression test has a size of ~250 nm with its two primary twin

boundaries parallel to the basal plane in the matrix, deviating away from the twinning plane {10$\bar{1}$2}. Therefore, the twin lamella captured here is not yet a fully "mature" twin, a term that describes the large microscopic twins typically characterized in post-mortem analyses of deformed Mg[29].

In compression testing of the TWP with the narrowest top width of 100 nm, in which the greatest stress difference is generated (Fig. 3a), twin formation and its tip movement can be tracked, as shown in Fig. 3 and Supplementary Fig. 8 (also see Supplementary Movie 3). Unlike the other TWP compression

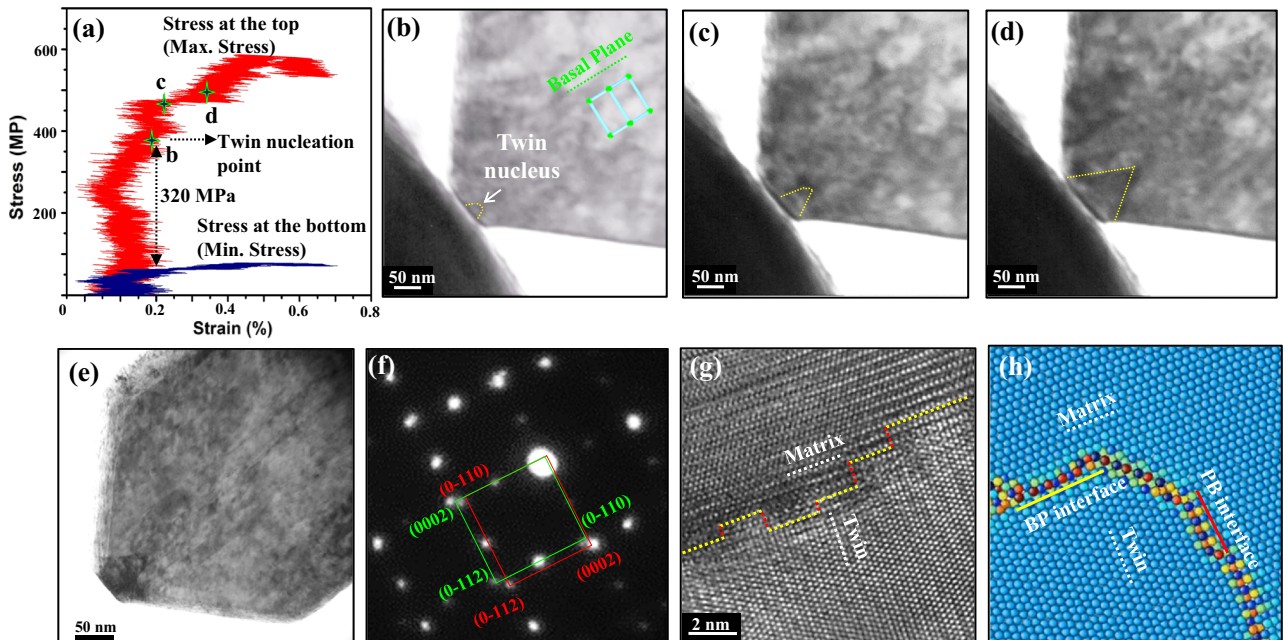

**Fig. 3 Twinning events in the truncated wedge-shaped Mg pillar with the 100 nm top in which there is a high stress difference. a** Corresponding compression curves of the pillar. The max. stress curve was calculated by the ratios of compression loading to the cross section of the pillar top. The min. stress curve was calculated by the ratios of compression loading to the cross section of the pillar bottom. The value of 320 MPa shows the stress gap at the twin nucleation point b. **b–d** Bright-field TEM snap-shots shows twin nucleation from the pillar top. The dashed yellow lines in **b–d** indicate the locations of twin boundaries. **e** Bright-field TEM image shows the morphology of the twin nucleus. **f** A selected area diffraction pattern demonstrates the orientation relationship between twin and matrix. The diffraction pattern is associated with a semi-coherent twin embryo structure where the coherency stress has been relaxed during sample thinning for HRTEM observation. **g** HRTEM images of twin boundaries shows the basal/prismatic terraced boundary. The yellow dashed lines indicate the basal-prismatic (BP) interfaces, and the red lines indicate the prismatic-basal (PB) interfaces. **h** Simulated atomic twin boundary structure with BP or PB interfaces. Atoms are color coded according to their excess energy, with the lowest energy atoms appearing blue, while the highest energy atoms appear red. The white dashed lines in **g**, **h** indicate the basal planes.

tests, the location where the embryo first forms can be detected, since its early-stage growth was confined as it attempts to expand out of the stressed region where there is less driving force to move. Here, we observe a triangular twin with a size of ~20 nm starting from the compression platen/nanopillar contact surface (Fig. 3b), when only 0.2% total strain is applied (Supplementary Fig. 3l). While, in the other two TWPs, the twin nucleation was first spotted at much larger total strain values (Supplementary Fig. 3). Together with the triangular shape of the twin, this may suggest that it is an embryonic one in the pillar with the 100 nm top. With continuous loading, this twin embryo expanded in size to be ~90 nm as measured from the TEM image in Fig. 3e. Diffraction pattern (Fig. 3f) and HRTEM (Fig. 3g) characterization of the boundary of the twin embryo reveals that it is comprised of basal-prismatic (BP) and prismatic-basal (PB) interfaces (BP denotes the interface parallel to basal plane in the matrix; PB denotes the interface parallel to prismatic plane in the matrix). The simulated twin nucleus with atomic BP and PB interfaces are shown in Fig. 3h.

## Discussion

**Twin formation and early-stage growth mechanisms.** We combined in-situ dynamic result and postmortem microscopy characterization to uncover information regarding the nucleation and growth mechanisms of a deformation twin nucleus. In the conventional pillar tests, there are two proposed early-stage twin growth mechanisms: shearing along a twinning plane and prismatic-basal transformation, as schematically shown in Supplementary Fig. 4. First, a narrow {10$\bar{1}$2} twin first initiates at one corner of the pillar and propagates its tip extremely fast across the width of the pillar (see Fig. 2a). The movement of the primary

twin boundary and tip indicate the growth mechanism. Since the twin boundary moves mostly along the (10$\bar{1}$2) twinning plane, this step is likely accomplished by conventional shearing-shuffle associated with successive gliding of twinning dislocations[6,11,30]. Once its twin tip reaches the other side of the pillars, it then mostly expands in thickness along the basal plane (Fig. 2b–d). We can characterize the crystallographic features of the boundaries of the twin nucleus after growing along the basal plane. TEM characterization indicates that the twin boundary (Fig. 2g) is serrated, comprised of BP and PB boundaries. Thus, thickening of the two twin boundaries along the basal plane likely occurs via the migration of BP and PB boundaries. These observations are in agreement with a recent TEM study of twin growth in conventional nano-pillars[27].

In the TWP with a top width of 400 nm, a similar two-step process for early-stage growth was observed, as schematically shown in Supplementary Fig. 4. Two {10$\bar{1}$2} twin variants nucleate from the two corners, then shear along the twinning plane to meet with each other at the middle of the pillar (Supplementary Fig. 3b). This step likely proceeds via the shear-shuffle growth. Then the primary boundaries migrate along the basal plane to form two co-zone twins (Fig. 1f). This step is accomplished via basal-prismatic transformation.

For the 250 nm and 100 nm TWPs, the nucleation and early-stage growth of a {10$\bar{1}$2} twin can be distinguished, and only one major early-stage growth mechanism can be identified, as no lateral shearing growth is detected, as schematically shown in Supplementary Fig. 4. One single triangular twin (~90 nm) was initiated from the top surface, and mainly move its primary boundary and tip along the basal plane in the upper region of the pillar, as shown in Fig. 2h–l (also see Supplementary Fig. 6). The

right boundary of the twin nucleus is parallel to the basal plane in the matrix, corresponding to a BP boundary (Fig. 2i). During compression, the BP boundary quickly migrates sideways (referred to as the B → P transformation), and then the twin nucleus propagates or grows along the basal plane (compression direction) (Fig. 2k). The primary boundaries are BP boundaries in the two sides of the twin (Fig. 2n). The inclined boundaries are mostly BP terraced interfaces where the basal plane in the matrix is parallel to the prismatic plane in the twin (Supplementary Fig. 7d, e), and the boundary of the twin tip is a PB terraced interface where the prismatic plane in the matrix is parallel to the basal plane in the twin (Fig. 2o). This twin is evidently still in its early stages of growth, since its boundary structure is distinguished from the "adult" $\{10\bar{1}2\}$ twin, where the primary boundary is parallel to the $\{10\bar{1}2\}$ twin plane.

It is worth noting that the dynamic processes associated with the P → B and B → P transformation occur at an atomic resolution much finer than can be accessed via in-situ TEM observation. This is mainly due to the "crystal size effect" of the samples[2]. Dynamic atomic imaging typically requires an extremely thin sample, within which the stress state, as well as deformation mechanisms, are different from those normally observed in bulk materials[2]. At such a small scale, for instance, deformation twinning is generally replaced by ordinary dislocation plasticity. Although twin nucleation was observed in rhenium nanocrystals with samples size well below 50 nm[31], in our in-situ experiments, twin formation was not observed in the Mg pillars with thicknesses of ~100 nm, and, instead, the pillars deformed via dislocations, as observed in Supplementary Fig. 9. In contrast, at the submicron scale, although the stress required for deformation twinning increases with decreasing sample size, compared to that of bulk materials[2], twins with BP and PB terraced boundaries were observed to form in all submicron pillars with a thickness of 750 nm. Stress variation in submicron samples is likely to alter the location of the twin nucleation, but not the mechanism by which a twin nucleates.

Thereby, a strategy to justify the P → B and B → P transformation mechanisms is to characterize twin boundaries and their migration, especially the early-stage twin tip movement, in the thick submicron pillars (750 nm in thickness) during the in-situ tests to avoid major sample size effects, and then do postmortem atomic-resolution TEM characterization on the formed twin nucleus after the sample has been thinned enough for subsequent atomic imaging. If the twin propagates along either a basal or prismatic plane, instead of the $\{10\bar{1}2\}$ shearing direction, this suggests that the twin migrates by the P → B or B → P transformation. In the submicron TWP with the narrowest top width of 100 nm, a small twin nucleus (~20 nm) is revealed to initiate from the top surface (Fig. 3b). The nucleus shows a triangular shape with its boundary and tip parallel to the basal plane, when it is first identified. Postmortem atomic TEM characterization (Fig. 3i) indicates the resulting twin nuclei are both composed of BP and PB terraced boundaries. Therefore, the combination of in-situ and postmortem observation provide evidence that the twin nucleates via the P → B and B → P transformation. To clearly identify the early-stage growth event of the twin nucleus, we characterized the twin boundary and tip movement using dark field TEM. As shown in Supplementary Fig. 10 and Supplementary Movie 4, once a twin nucleates from the top surface, it moves its tip and boundary along the basal plane (compression direction) under further compression, and no lateral shearing along the twin plane is detected. This suggests that the crystal geometry confines the twin propagation along the twinning plane and enables the early-stage growth along basal plane via the P → B and B → P transformation.

## Simulation validation of pure-shuffle twin nucleation.
Thus far, all in-situ TEM observations here reveal that the twin nucleation and early-stage growth events involve the creation of PB and BP twin boundaries. Their existence indicates that the nucleation stages occur via Prismatic ↔ Basal transformation mechanisms, termed as a "pure-shuffle" nucleation mechanism. Several theoretical and simulation studies, performed at different scales, have proposed that a deformation twin propagates and grows via nucleation and glide of twinning dislocations/disconnections on the twin plane under the shear stress along the twin direction, associated with atomic shuffling events (referred to as shear-shuffle mechanisms)[19,30–34]. A major controversy concerning the description of $\{10\bar{1}2\}$ twinning (the most common type in HCP metals) has recently developed principally based on atomistic simulations[27,32,33]. The debate is focused on whether twinning follows a conventional shear-shuffle nucleation mechanism (such as the pole mechanism involving twinning dislocations) or a recently proposed pure-shuffle nucleation mechanism[6,11]. Our observations support the latter for $\{10\bar{1}2\}$ twinning.

To understand the atomic-level process underlying the in-situ observations, we carried out MD simulations for the nucleation and early-stage growth events (see Fig. 4, Supplementary Discussion and Supplementary Fig. 11).

Figure 4a (0 ps) shows the relaxed wedge-shaped pillar. Figure 4b (84 ps) shows the nucleation of a twin nucleus with 2–3 layers via B → P transformation at the left corner and one layer twin nucleus at the right corner by elastic relaxation due to free surfaces. This mainly results in PB interfaces parallel to the basal plane of the twin nucleus, as indicated by the green dashed line in the Fig. 4b. At the next moment, the left twin nucleus expands sideways to become a 4-layer twin nucleus mainly via B → P transformation, and its boundary is mainly composed by PB interfaces (Fig. 4c). The one-layer nucleus seems to be unstable and disappears at the right corner via a reversed P → B transformation (Fig. 4c). With continuous compression strain, the twin nucleus at the left side continuously grows layer-by-layer sideways via B → P transformation and vertically via P → B transformation. Its boundary is still mainly composed of PB interfaces. Also, another twin nucleus with two-layer PB interfaces initiates from the right corner (Fig. 4d). Immediately, with continuous loading along the basal plane, this nucleus now can grow its boundary layer-by-layer to form a triangle-like nucleus with multiple layers of PB and BP interfaces (Fig. 4e, f). The one-layer nucleus is not stable in a Mg crystal, and a minimum, stable nucleus involves a thickness of at least several crystallographic planes[6,11]. A massive pure-shuffle mechanism is thus required to create a stable volume with multiple layers to compete with a structural reversal, or detwinning.

The subsequent coalescence of twin nuclei can result in a more stable twin nucleus with more volume, as shown in Fig. 4d–g (90–102 ps). The two twin nuclei at the corners of the truncated wedge-shaped model have the same variant (the same orientation). Once they meet, they are favored to accommodate their boundaries via the P → B and B → P transformation. Eventually, a larger triangle-like twin nucleus is formed, as shown in Fig. 4h. This morphology agrees with the in-situ TEM observation at the moments when the twin nuclei were first identified in the TWPs (Figs. 2h and 3b). The early-stage growth of the triangle-like twin is shown in Fig. 4h, i (106–140 ps). During this process, the twin nucleus moves its boundary through the migration of BP and PB interfaces. In agreement with the experimental observation (Fig. 2h–k and Supplementary Fig. 6), the twin nucleus evolves its boundary from a triangle-like shape to a truncated wedge shape, due to the geometric confinement. The twin nucleus propagates and grows sideways via the B → P transformation and

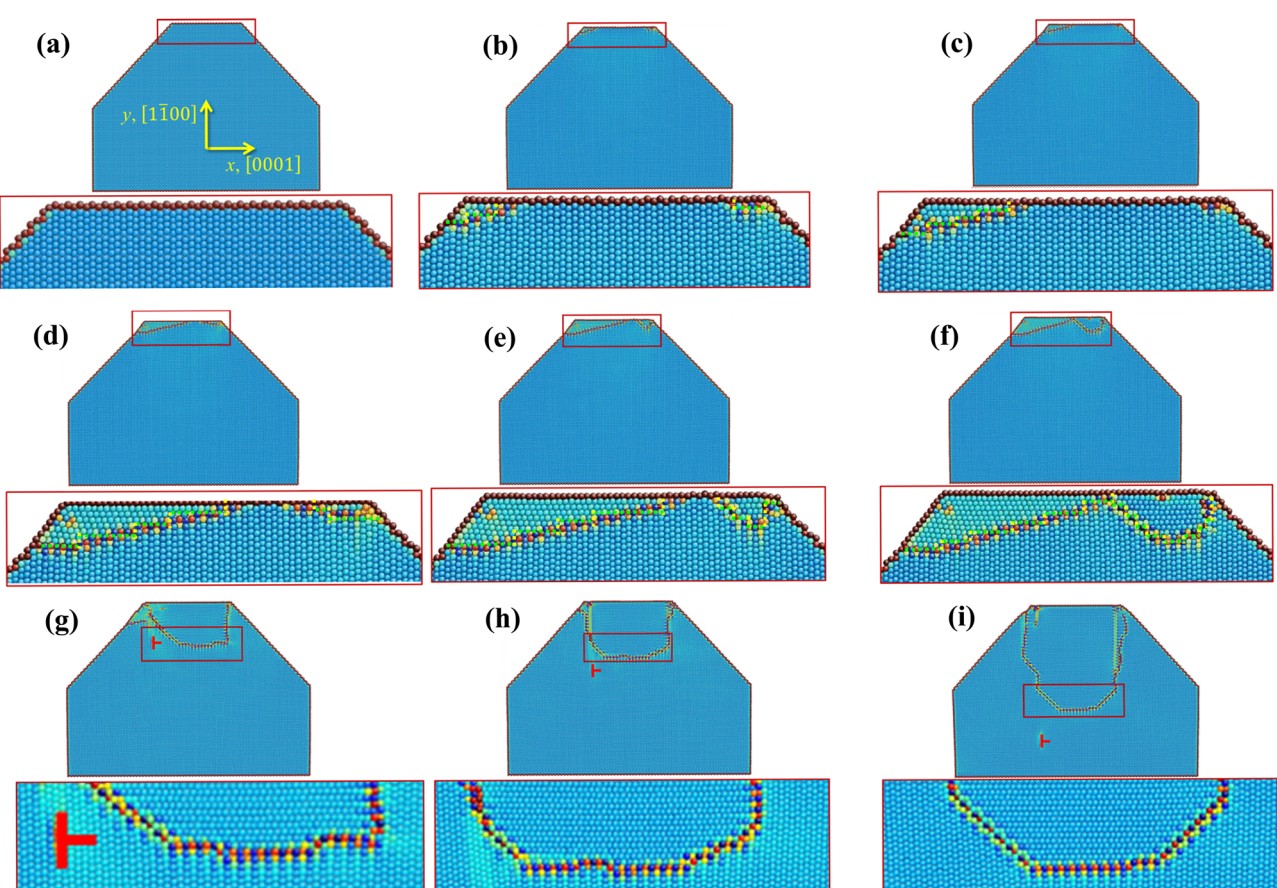

**Fig. 4 Molecular dynamic simulations of twin nucleation. a** Initial relaxed single crystal. **b–d** Nucleation of two twin nuclei at the corners on top surface. **e–g** Merge of two twin nuclei into a larger twin nucleus. **h, i** Propagation and growth of twin nucleus via migration of basal-prismatic (BP) and prismatic-basal (PB) interfaces. The symbol "T" indicates a basal <a> dislocation that is emitted from the PB interface. The bottom images of **a–i** are magnified images in the pillar top region. The rectangles in each image indicate locations associated with the enlarged images. The twin nucleus is surrounded by PB and BP interfaces. In the enlarged images, the yellow dashed lines indicate the BP interfaces, and the green dashed lines indicate the PB interfaces. Atoms are color coded according to their excess energy, with the lowest energy atoms appearing blue, while the highest energy atoms appear red.

vertically along the normal of the prismatic plane in the matrix via the P → B transformation. The truncated wedge-shaped twin boundaries are primarily composed of BP interfaces, as seen in both the experimental (Fig. 2n, o and Supplementary Fig. 7) and MD (Fig. 4i) results. This suggests that the vertical migration of the twin boundary via P → B transformation dominates the early growth stage to form the truncated wedge shape. More details on the MD results are described in Supplementary Fig. 11 and Supplementary Movie 5.

As in the experiment, the twin embryo is surrounded by a series of short, connected PB and BP boundaries. Contrary to conventional thought, no part of the embryo boundary is a coherent twin boundary (CTB), which is the conventional boundary seen in fully developed twins. Several prior atomic-scale calculations have shown that the long CTBs of conventional twins are formed by gliding twinning dislocations[29]. To identify the mechanism for forming BP and PB facets, we constructed the strained, coherent dichromatic complex, where the lattices of the two crystals are coherent, denoted by the green rectangle in Fig. 5b. The green arrows are shuffle vectors associated with one unit transformation. Their summation is equal to zero. Unlike CTBs, the BP and PB facets form by atomic shuffling.

To rationalize the experimental and MD simulations, we can estimate the work done in forming a fully PB and BP bounded nucleus, which would require pure atomic shuffling, versus forming a nucleus bounded partially by PB, BP, and CTBs, the latter of which forms via glide of twinning dislocations plus atomic shuffling. When a twin embryo first nucleates within the crystal, the embryo is small enough that the twin boundary will be coherent. The surface-to-volume ratio varies as $1/<r>$, where $<r>$ is the mean embryo radius. For small $<r>$, the surface term dominates, and the lower interfacial energy of the coherent interfaces, compared to that for a semi-coherent boundary with misfit dislocations, causes the initial twin nucleus to be coherent in the matrix, as is common for all nucleation processes[34,35]. Using the molecular statics (MS) method, we further calculated the interface formation energy of CTB, coherent PB (CPB) and coherent BP (CBP) boundaries. For Mg, the MS method finds that the CPB or CBP boundary has the lower formation energy ($\gamma_{CTB} = 105 \frac{mJ}{m^2}$) than CTB ($\gamma_{CTB} = 125 \frac{mJ}{m^2}$). This explains why the embryo is surrounded by CPB, CBP boundaries and not CTB boundaries (see Fig. 5c).

To complete the explanation, we conduct a mechanics analysis to estimate the mechanical work involved in creating a twin nucleus via the two scenarios in sequence. The first scenario considers the work associated with P → B and B → P transformations (Fig. 5d), and the other competing scenario is associated with twinning shear on the twin plane (Fig. 5e). Disregarding the excess energy at the corners, the mechanical work done for forming each twin can be calculated by the summation of boundary energies and the strain energy stored in the nucleus. For the first case, i.e., the twin nucleus shown in Fig. 5d, the mechanical work per unit

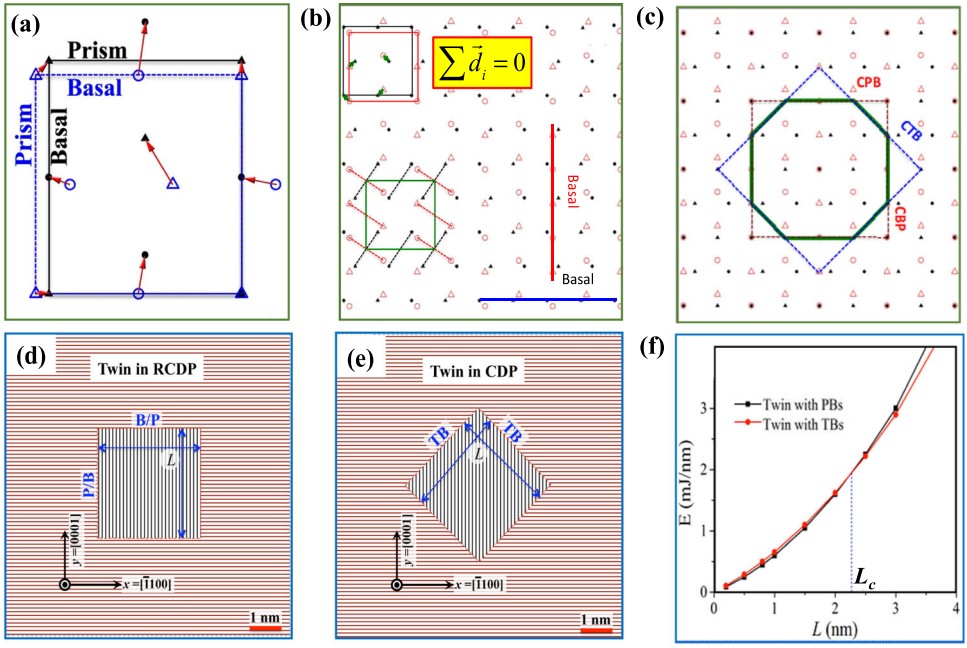

**Fig. 5 Pure-shuffle nucleation mechanisms. a** Two unit cells with a 90° rotation showing lattice mismatch. The blue dashed square and the black square indicate the unit cell. The circles and the triangles represent the positions of atoms and the red arrows indicate the shuffle directions. **b** Strained coherent dichromatic complex associated with the 90° rotation of two crystals. The green rectangle represents the coherent lattice, and the red and black rectangles represent the two crystals. The green arrows indicate the shuffle vectors and their summation is equal to zero. Both matrix and embryo have been equally strained to form coherent lattice and complex structures. **c** The modified strained coherent dichromatic complex by overlapping one set of atom sites, showing possible boundaries associated with a twin nucleus. CTB represents a coherent twin boundary, CBP represents a coherent basal-prismatic (BP) boundary, and CPB represents a coherent prismatic-basal (PB) boundary. The CPB or CBP deviates 1.9 degrees from the zero-stress CTB. **d, e** Schematics of two twin nuclei with various twin boundaries. CDP indicates coherent dichromatic pattern and RCDP indicates rotation coherent dichromatic pattern. **f** variation in mechanical work as a function of the twin dimension.

thickness is equal to $W_{CPB} = 4L\gamma_{CPB} + \left(\frac{1}{2}\sigma_{xx}\varepsilon_{xx} + \frac{1}{2}\sigma_{yy}\varepsilon_{yy}\right)L^2$. For the second case, i.e., the twin shown in Fig. 5e, $W_{CTB} = 4L\gamma_{CTB} + (\tau_T\varepsilon_T)L^2$. For Mg, the lattice mismatch between the basal and prismatic plane is 6.5% (Fig. 5a).

Prismatic ↔ Basal transformations result in an extension strain $\varepsilon_{xx} = 6.5\%$ in one direction (BP) and a contraction strain $\varepsilon_{yy} = -6.5\%$ in the orthogonal direction (PB), and together they are equivalent to shear strain associated with the twinning shear, $\varepsilon_T$. Using the effective anisotropic elastic stiffness tensor of pure Mg ($C_{11} = 67.5$ GPa, $C_{22} = 61.8$ GPa, $C_{33} = 61.8$ GPa, $C_{44} = 18.2$ GPa, $C_{55} = 19.9$ GPa, $C_{66} = 18.2$ GPa, where the x-axis is along [0001], the y-axis is along $[10\bar{1}0]$, and the z-axis is along $[1\bar{2}10]$), we compute the variation in the two mechanical work terms for a twin dimension, $L$. The calculation ignores the contribution of the elastic energy due to the compressive stress, since there is a slight difference between the elastic modulus along the normal of the basal and prismatic planes, 67.5 GPa and 61.8 GPa, respectively, and no change in shear modulus along the twin plane. Figure 5f shows that the twin surrounded by CPB and CBP boundaries requires a lower mechanical work than the twin associated with twinning shear when the twin dimension is smaller than 2.5 nm. When an anisotropic elastic stiffness tensor is used, the energy difference originates due to the difference in interface energy. Although the missing component in this simple analysis is the excess energy associated with corners and their interaction, it still indicates that twin nucleation via P → B and B → P transformations is energetically more favorable than nucleation via glide of twinning dislocations. In this case, the stresses needed for nucleation can be identified. For the nucleation from the pillar upper surface, the P → B transformation is driven by normal

compression while the B → P transformation is accommodated by elastic relaxation due to free surfaces. A twin nucleus is thus formed.

Next, the initial growth of this twin nucleus is in question. This newly formed (nano-sized) twin embryo is initially bounded by CPB and CBP interfaces, and undeniably with attendant large coherency stresses due to the lattice mismatch of 6.5% across these interfaces. Expansion of this embryo by the migration and lengthening of the PB and BP interfaces would be accompanied by prohibitively increasing elastic strain energy in the matrix. However, these coherency stresses can be relaxed by the formation of misfit dislocations on PB and BP boundaries, enabling these boundaries to lengthen and hence the embryo to grow. Geometric analysis of dislocation reactions identifies that misfit dislocations for the PB and BP interfaces can form via the nucleation and emission of basal <a> dislocations. As evidence, TEM analysis was carried out at the growing twin boundary and TEM images (see Supplementary Fig. 12) clearly show a significant number of <a> dislocations in the front of the growing twin. In addition, MD simulations reproduce the emission of one basal <a> dislocation from a PB interface (Fig. 4). The emission produces one <a> dislocation on the basal plane gliding into the matrix (Fig. 4h–j) and one residual partial dislocation, as a misfit dislocation to balance the lattice mismatch on the PB interface. The accumulated <a> dislocations within the nano-sized single crystal sample can be driven out of the sample by the image forces, which further facilitates the formation of more PB and BP interfaces[6]. This analysis explains how the twin boundary of the embryo, in its initial stages of growth, becomes comprised of long PB and BP semi-coherent interfaces with misfit dislocations. Initial embryo growth is thereby accomplished

through the migration of the PB or BP interfaces associated with nucleation and propagation of basal <a> dislocations.

This work has revealed the transformation of the stress-induced embryo with a CPB/CBP boundary to a growing embryo with a terraced, semi-coherent BP/PB boundary-structures that are unlike fully developed $\{10\bar{1}2\}$ twins frequently reported in the literature. The key question then becomes how a twin nucleus eventually grows to be an "adult" $\{10\bar{1}2\}$ twin, frequently seen in deformed samples Mg, with its long planar $\{10\bar{1}2\}$ CTBs.

First, there is an orientation difference between an embryo with the CPB/CBP boundary (90° between twin and matrix) and an "adult" $\{10\bar{1}2\}$ twin (86.22° between twin and matrix). An additional small rotation of the embryo into the twin orientation is needed to form a mature twin[11,33]. For an "adult" twin boundary, the rotation of 86.22° is equally partitioned between the twin and matrix[11,32]. When a twin first nucleates from the matrix, the rotation is not equally partitioned between matrix and twin due to the stiffness of the matrix. The rotation (3.78° derived from 90°) is imposed entirely on the embryo and can be accommodated at the CPB and CBP interfaces. When a twin embryo grows, partitioning of the rotation into the matrix and twin occurs, thus creating the true twin interfaces[11]the stiffness of the matrix

In addition, we performed MD simulation and topological analysis for initial growth of a twin nucleus surrounded by PB and BP boundaries (see Supplementary Fig. 13a and Supplementary Movie 6). The topological character of PB and BP facets has been well summarized in the references[6,30,36,37]. The PB and BP steps/facets have dislocation character and the misfit on the interface, have features of a coherency disclination[6,37]. Analogous to the Frank analysis for crystal growth[6], we observe in calculation that the PB and BP facets grow faster (see Supplementary Fig. 13b, c) than the $\{10\bar{1}2\}$ CTB during twin nucleation. As the twin nucleus expands, the misfit dislocations that must nucleate on the PB and BP facets to release the coherency stress also, at the same time, further pin the PB and BP boundaries (see Supplementary Fig. 13d, e). This impedes migration of PB and BP facets and leaves the coherent twin bounded by slower growing $\{10\bar{1}2\}$ twin planes. As a result, the growing twin forms a lenticular shape, Supplementary Fig. 13f, bounded by CTB terraces. This shape is consistent with the mature twin commonly found in bulk samples.

**Importance of the work**. Deformation twinning renders metals, such as Mg and other HCP metals, more unpredictable to design and use in critical load bearing applications, such as those that currently use steel and Al alloys. Understanding twinning mechanisms, especially early-stage growth mechanisms, will enable researchers to develop strategies for restricting or promoting twinning in materials. We found that the early-stage formation of a twin nucleus associated with instability of plastic flow can be controlled by tailoring the sample geometry. As seen in Supplementary Fig. 3, there are strain bursts around the twin nucleation events, as indicated by the horizonal arrows in the compression curves. The magnitude of the strain bursts is determined by the twin nucleation and early-stage growth. The smaller the wedge top of the pillars used, the earlier the twin nucleated, and the smaller the strain burst observed. In the pillars with a regular sample geometry, the early-stage growth of the twin nucleus was initiated by fast shearing along the twinning plane (Fig. 2a and Supplementary Movie 1). This typically results in unstable plastic flow with an abrupt strain burst (more than 1.2%), as seen in Supplementary Fig. 3i. This is associated with the highly anisotropic plasticity response in bulk Mg, making it

too unreliable for structural use. In contrast, in strategically designed TWPs, the abrupt strain burst is greatly suppressed with decreasing pillar top width. In TWPs with a top width of 400 nm, ~0.4% strain burst (Supplementary Fig. 3j) is associated with the twin nucleation and its early-stage growth. In the TWPs with top widths of 250 nm and 100 nm, ~0.2% strain burst is spotted at the moment when the twin nucleus is first identified, as shown in Fig. S3k–l. This is because the twinning related early-stage deformation is dominated by the slower migration of BP and PB interfaces layer-by-layer due to the crystal geometry confinement. Therefore, a more stable and continuous plastic flow can be accomplished, which suggests a potential strategy to enhance the plasticity of Mg via the geometric design of the grains in polycrystalline Mg. It is also worth noting here that twin propagation can be impeded by the interaction of twins within the designed geometry. The "obstacle effect" posed by twin–twin interactions slow their propagation, leading to stress bursts in the compression curve (Supplementary Fig. 3j), implying strain hardening via crystal geometry confinement.

In summary, we have studied the effects of stress field differences on twinning in single-crystal pillars with strategically designed geometries, allowing observation of nucleation and early-stage growth mechanisms of $\{10\bar{1}2\}$ deformation twins. The captured twin nucleus is characterized to be surrounded by prismatic-basal and basal-prismatic boundaries and not the often seen conventional $\{10\bar{1}2\}$ coherent twin boundaries of mature twins. In-situ TEM analysis and atomic-scale simulation indicate that these earliest stages of $\{10\bar{1}2\}$ twin embryo formation occur via prismatic-to-basal and basal-to-prismatic transformations, a mechanism that requires pure atomic-shuffle, with zero net shear or displacement along the twinning plane. Moreover, of significance is the verification in a crystal geometry dependent growth mechanism of the twin nucleus at the early stage. The conventional growth mechanism along the twinning plane via glide-shuffle is proposed to be associated with the instability of the plastic flow in the pillars with a regular shape, whereas early-stage growth accomplished via the gradual migration of PB and BP steps can accommodate a more stable plastic flow in the strategically designed TWPs. This suggests a potential strategy to enhance plasticity via the crystal geometry confinement. The nucleation and early-stage growth model proposed here is not only particular to $\{10\bar{1}2\}$ in Mg, but also can be generalized to $\{10\bar{1}2\}$ twins in most of the HCP metals. Also, the method we developed here can be readily used for studies of nucleation and growth mechanisms for twinning and phase transformations that commonly occur in other crystalline materials.

## Methods

**In-situ TEM compression experiment**. The single-crystal Mg pillars are prepared by focus ion beam machining from a pure single-crystal Mg cylinder with a 3 mm diameter. A Mg single crystal with (0001) orientation (purity 99.999%) was purchased from Goodfellow (Coraopolis, PA, USA). The single crystal was first mechanically polished, with a final mechanical step of 50 nm $SiO_2$ slurry in a diluted solution, then etched in order to remove the damage layer and reveal the presence of any twins. Compression perpendicular to the (0001) axis was used to introduce extension twins in the crystal. A slice of ~500 µm was sectioned from the single crystal using a wire saw with minimal speed and load. Multiple H-bar type foils (20 µm in length, 10 µm in width, 750 nm in thickness), as shown in Supplementary Fig. 1a, were prepared from the slice by using focused ion beam (FIB) machining on a FEI Scios 3-D dual beam (SEM/FIB) system (Hillsboro, OR). Next, pillars with different geometries (Supplementary Fig. 1) were prepared by the FIB micromachining. The pillars have a thickness of 750 nm and height of 1200 nm (Supplementary Fig. 1d). The thick pillars are designed to avoid major crystal size effects. The details on the samples are listed in Supplementary Table 1.

In this work, we used the following approach to minimize the presence of artifacts that may be introduced by the FIB. First, we fabricated samples using cross section cutting on the lateral surface. Second, a low-energy (2 keV) ion beam was used to clean the sample surface and minimize the thickness of the damage layer.

TEM observation confirmed that the FIB affected layer was usually less than a few nanometers. This is negligible compared to the sample thickness of 750 nm.

In-situ mechanical testing was conducted on the single-crystal pillars using a Hysitron PicoIndenter (PI95) inside a JEOL 3010FEG TEM (300 kV) and TEAM 1 TEM (a modified Thermo Fisher Scientific Titan TEM) equipped with double-aberration-corrector and high-performance K2-IS in-situ camera to capture twin nucleation. The in-situ compression test was conducted using the compression platen with 1μm width at the top to compress the pillars, as shown in Supplementary Fig. 1c, with the indentation direction being parallel to the basal plane. The indentation speed was 0.5 nm/s for the rectangular pillars and the TWPs with the sub-micrometer tops (400 nm, 250 nm). For the TWPs with the nano-sized top (100 nm), the twin was typically generated much earlier; thus, a lower indentation speed (0.1 nm/s) was then used to capture more details of the twinning process. All of the in-situ tests were carefully aligned using the electron beam to avoid any misalignment between the compression platen and pillars. Details of the experimental conditions used in our study are listed in Supplementary Table 1. The corresponding in-situ compression curves are shown in Supplementary Fig. 3.

To investigate the atomic structure of the twin embryos, the deformed samples with thickness of 750 nm were further thinned by using FIB and a Nano Mill (Fischione Inc). The TEM/STEM images were acquired using a 200 kV JOEL-2800 TEM and a Thermo Fisher Scientific Titan TEM with double-aberration-corrector under 300 kV imaging mode.

**Simulations**. Using the Finite Element Method with an Abaqus/CAE® solver, we modeled the stress fields generated in the pillar due to compressive loading applied to the upper surface of the pillar (Supplementary Fig. 2). The regular pillar model and three TWP models with top widths of 400, 250, and 100 nm are meshed with hexahedron elements. Models are then assigned materials properties of Mg. For all models, x-axis is along [0001] direction, y-axis is along $[\bar{1}010]$ direction and z-axis is along $[1\bar{2}10]$ direction. A 5 nm displacement is applied to the top surfaces while the bottom surfaces are fixed. In post analysis, we show stresses, $\sigma_b$, $\sigma_p$, and $\tau_{tw}$ on the middle plane in the thickness direction. The normal stresses in the x-direction are denoted as $\sigma_b$ as shown in Supplementary Fig. 2a–d and in the y-direction as $\sigma_p$ in Supplementary Fig. 2e–h. $\tau_{tw}$ (Supplementary Fig. 2i–l) is defined as the shear stress on the two 45° inclined planes (corresponding to two twin planes).

Atomistic simulations were conducted by using the commercially available software, Large-scale Atomic/Molecular Massively Parallel Simulator[38] (https://www.lammps.org). Atomistic simulations were performed for Mg using empirical interatomic potential developed by Liu et al.[39] to explore the twin nucleation process. A recent study indicates that both Liu's potential and Sun's potential can accurately reproduce the formation energies of twin boundary and PB interface[40]. However, Liu's potential is much closer to the density functional theory (DFT) calculations of surface energies, and also recreates the dependence of surface energy on normalized atom density, while the Sun potential struggles to recreate the DFT values[40]. Therefore, Liu's potential was chosen in this work. A TWP is created so that the x-axis is along the [0001] direction, the y-axis is along the $[\bar{1}010]$ direction, and the z-axis is along the $[1\bar{2}10]$ direction. The relaxed TWP has a top width of 12 nm, a bottom width of 40 nm, and a height of 30 nm. An indentation compressive load is applied on the top surface of the model at a displacement rate of 10 m/s at a temperature of 300 K. Periodic boundary conditions were adopted in the z-direction, while a 1 nm thick region at the bottom of the model is fixed. The periodic thickness in the z-direction is 3.0 nm. The twin nucleus is therefore essentially straight along the z-direction due to the small sample thickness and periodic boundary conditions. Usage of a smaller sample size in one direction is a practical choice for MD simulations to allow for larger sizes in the other directions to capture the important physics. In addition, such a quasi-2D shape is analogous to the experimental setup where the TEM sample is 750 nm thick along the z-direction and the indenter tip is 1000 nm. In addition, our experiments show a clear twin nucleus through the sample thickness.

## Data availability
All data generated or analyzed during this study are included in this article (and its Supplementary information files).

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

## Acknowledgements

The authors acknowledge financial support from the National Science Foundation (NSF-CMMI-1631873, NSF CMMI-1729829, and NSF CMMI-1723539). We also acknowledge funding from the ONR–Defense University Research Instrumentation Program under grants N00014-13-1-0668 and N00014-11-1-0788, which supported the purchases of the dual-beam FIB/SEM and the Hysitron Picocompression platen, respectively. I.J.B. acknowledges financial support from the National Science Foundation (NSF-CMMI-1729887). Work at the Molecular Foundry was supported by the Office of Science, Office of Basic Energy Sciences, of the U.S. Department of Energy under Contract No. DE-AC02-05CH11231. The work at LLNL was performed under the auspices of the U.S. Department of Energy under contract No. DE-AC52-07NA27344. Y.M.W. was supported by the National Science Foundation (NSF-DMR-2104933). M.G. and J.W. acknowledge financial support from the National Science Foundation (NSF-CMMI-1661686). Atomistic simulations were completed utilizing the Holland Computing Center of the University of Nebraska, which receives support from the Nebraska Research Initiative. Some of the TEM experiments were conducted using the facilities in the Irvine Materials Research Institute (IMRI) at the University of California, Irvine.

## Author contributions

L.J., A.M., I.J.B., S.M., E.J.L. and J.M.S. conceived the research. L.J., J.W., I.J.B., and Y.M.W. wrote the manuscript. L.J., X.W., D.Z., J.C. and M.X. conducted the experiments. J.W., M.G., Z.P. and T.R. conducted the simulation. L.J., Y.M.W., J.W., I.J.B., X.P., E.J.L. and J.M.S. analyzed and interpreted the data. All authors reviewed the manuscript.

## Competing interests

The authors declare no competing interests.
