## [Peer Review File · Nature Communications]

Title: Visualization and Validation of Twin Nucleation and Early-stage Growth in MagnesiumREVIEWER COMMENTS

Reviewer #1 (Remarks to the Author):

This paper describes a combined experimental and theoretical study of the nucleation and early-stage growth of {10-12} twins in Mg. {10-12} is the most prolific twinning system in deformed Mg and contributes to problematic plastic anisotropy encountered in the manufacture of Mg components. Nucleation and early-stage growth of twins is difficult to investigate by conventional experimental means, but the authors have devised a clever technique to overcome these difficulties. 750nm square pillars are obtained by focussed-ion beam machining, and the widths of their top surfaces have been reduced to 400nm, 250nm and 100nm, forming a series of tapered wedge pillars (TWPs). When these are compressed in-situ by the platen of a nano-indenter, nucleation and early stage twin growth is observed in the narrowest TWPs because the inhomogeneous stress distribution induced is sufficient for nucleation but insufficient for subsequent growth into mature twins. The atomic structure of such twin embryos has then been investigated after unloading and further specimen thinning by HRTEM. In parallel with the experimental programme, atomic-scale computer simulation is used to study a nucleation mechanism involving atomic shuffling, as opposed to the classical model which invokes a shear-shuffle mechanism by dislocation motion along coherent twin boundaries.

The authors conclude that nucleation occurs by the atomic shuffling mechanism, thereby forming twin embryos bounded by configurations of basal-prism (BP) and prism-basal (PB) interfaces as distinct from the classical model involving coherent twin boundaries (CTBs). Furthermore, they propose that early-stage growth proceeds by migration of the BP/PB boundary configurations.

This paper reports an outstanding study of an important technological issue, and eventual publication is recommended. However, some aspects of the current text require attention, as summarised below.

1. In fig. 1a the twin induced in the 750nm pillar (no wedge) appears to have the conventional lenticular form based on {10-12} CTBs. However, the HRTEM image in fig.1g shows that the interface is actually comprised of PB/BP segments. Is this observation inconsistent with the authors' conclusion?
2. Fig. 3 relates to the 100nm TWP. Fig. 3f shows that the interface bounding the embryo is an array of PB/BP segments. However, the diffraction pattern in fig.3f shows that the matrix and embryo crystals are fully relaxed. Has the misfit in such a small embryo already been relieved?
3. Can the authors explain what they mean by "geometry confinement" (e.g. in fig.2b-d)?
4. The simulation images in fig.4 need to be enlarged in order that the embryo boundary structure can be discerned by the reader. Is this interface coherent?
5. Have both the matrix and embryo been strained equally in fig. 5b? What is the meaning of CTB in such a bicrystal?
6. There is no analysis of the topological character of PB/BP "steps": do they have dislocation character? Can they move laterally without diffusion? What defects are required to accommodate misfit? Would the presence of misfit dislocations impede migration of PB/BP boundaries?
7. Throughout the text, the orientation relationship between the matrix and embryo is 90° about $\langle 1-210 \rangle$. How does the additional small rotation about this axis arise in order that a true (mature) twin is formed?

R C Pond

Reviewer #2 (Remarks to the Author):

This paper presents a breakthrough in visualization and validation of a twinning mechanism in magnesium, which has been proposed but unverified for the past 50 years. The most innovative design for experimental validation is the science-guided truncated wedge-shaped pillars (TWPs) from single-crystal Mg, which controls a steep stress field in the crystal under compression that can isolate twin nucleation from twin growth. With the TWPs, atomistic simulations and topological analysis were applied for a systematic study on the mechanism. The paper is well organized and professionally written. The microstructure characterization is excellent.

In the section of "Importance of the work", the authors made it clear that the advance in understanding twinning mechanisms, especially early stage growth mechanisms, is of engineering significance to develop novel methods in controlling/tuning twinning for materials. For micro/nano components, this can be straightforward. For macroscopic parts, it might not be easy to implement the geometry design strategy for stress control at nanoscale. It seems that the innovative method developed in this work does provide a direct way to study the alloying effect on twin nucleation/initiation in Mg, thus facilitating the discovery of high performance Mg alloys for engineering significance.

The methodology in this work can be widely used for other twinning mechanisms in any other solid materials. Moreover, it can have a broad impact in study of nanoscale deformation mechanisms. It is a paper worthy of publication as a highlight in Nature Communications.

RESPONSE TO THE REVIEWERS

Title: Visualization and Validation of Twin Nucleation and Early-stage Growth in Magnesium

Authors: Jiang, Gong, Wang, Pan, Wang, Zhang, Wang, Ciston, Minor, Xu, Pan, Rupert,
Mahajan, Lavernia, Beyerlein, Schoenung

Reviewer #1 (Remarks to the Author):

This paper describes a combined experimental and theoretical study of the nucleation and early-stage growth of {10-12} twins in Mg. {10-12} is the most prolific twinning system in deformed Mg and contributes to problematic plastic anisotropy encountered in the manufacture of Mg components. Nucleation and early-stage growth of twins is difficult to investigate by conventional experimental means, but the authors have devised a clever technique to overcome these difficulties. 750nm square pillars are obtained by focussed-ion beam machining, and the widths of their top surfaces have been reduced to 400nm, 250nm and 100nm, forming a series of tapered wedge pillars (TWPs). When these are compressed in-situ by the platen of a nano-indenter, nucleation and early stage twin growth is observed in the narrowest TWPs because the inhomogeneous stress distribution induced is sufficient for nucleation but insufficient for subsequent growth into mature twins. The atomic structure of such twin embryos has then been investigated after unloading and further specimen thinning by HRTEM. In parallel with the experimental programme, atomic-scale computer simulation is used to study a nucleation mechanism involving atomic shuffling, as opposed to the classical model which invokes a shear-shuffle mechanism by dislocation motion along coherent twin boundaries.

The authors conclude that nucleation occurs by the atomic shuffling mechanism, thereby forming twin embryos bounded by configurations of basal-prism (BP) and prism-basal (PB) interfaces as distinct from the classical model involving coherent twin boundaries (CTBs). Furthermore, they propose that early-stage growth proceeds by migration of the BP/PB boundary configurations. This paper reports an outstanding study of an important technological issue, and eventual publication is recommended. However, some aspects of the current text require attention, as summarized below.

R C Pond

Response: We highly appreciate the reviewer (Prof. R C Pond) for recognizing the contribution of this work to understanding twin nucleation in hexagonal close packed (HCP) metals. We also thank Prof. R C Pond for the constructive comments and suggestions. We have revised the manuscript accordingly. All of the changes are highlighted in yellow in the revised manuscript.

1. In fig. 1a the twin induced in the 750nm pillar (no wedge) appears to have the conventional lenticular form based on {10-12} CTBs. However, the HRTEM image in fig.1g shows that the interface is actually comprised of PB/BP segments. Is this observation inconsistent with the authors' conclusion?

Response: The major conclusion of this work is that the nucleation of {10-12} twins is dominated by PB/BP interfaces associated with the pure-shuffling mechanism. We do not question the shear-shuffling mechanism that twinning disconnections glide on CTBs associated with atomic shuffling

(RC Pond, JP Hirth, A Serra and DJ Bacon, Mater Res Lett 4:4 185-190 (2016)) for twin propagation and growth under shear stress. However, in the nanoscale sample, twin growth could be accomplished by migration of prismatic-basal interfaces (Liu B-Y, et, al. Nature Communications 5, 3297 (2014); Gong, Wang, et. al, Materials Research Letters 5, 449-464 (2017)), since the misfit strain of the BP interface can be easily released by emission of basal dislocations. We have modified the manuscript to include a citation to this reference: “These observations are in agreement with a recent TEM study of twin growth in conventional nano-pillars²⁷”.

In the conventional 750 nm pillar (no wedge), a twin nucleates suddenly and immediately propagates across the pillar. Thus, we cannot characterize the early stage or initial stage of this twin nucleus. We can only analyze propagation and growth of a twin in a nanoscale sample. The PB/BP segments along the twin boundary are associated with propagation and growth of the twin in the nanoscale sample. Therefore, this observation of PB/BP segments along the TB does not conflict with our conclusion on twin nucleation.

2. Fig. 3 relates to the 100 nm TWP. Fig. 3f shows that the interface bounding the embryo is an array of PB/BP segments. However, the diffraction pattern in fig.3f shows that the matrix and embryo crystals are fully relaxed. Has the misfit in such a small embryo already been relieved?

Response: In principle, the initial embryo formed by the solid-state nucleation should have coherent PB and BP interfaces according to the pure-shuffling mechanism, as proposed by J Wang et al (Materials Research Letters, 1, 126-132 (2013)) and further pointed out by RC Pond et al (Materials Research Letters, 4:4, 185-190 (2016)). In this way, the matrix and twin embryo should show overlapping diffraction patterns. In reality, the coherency stress might be relaxed to form a

semi-coherent structure after unloading the sample or during thinning of the TEM samples. Thereby, Fig. 3f shows two separate diffraction patterns. In the revision, we have clarified this in the Fig.3f caption, as follows: “The diffraction pattern is associated with a semi-coherent twin embryo structure where the coherency stress has been relaxed during sample thinning for HRTEM observation.”

3. Can the authors explain what they mean by “geometry confinement” (e.g. in fig.2b-d)?

Response: The “geometry confinement” for the case of the conventional pillar is related to the large contact area and the vertical displacement loading (vertical to the contact surface of the pillar), both of which prevent the horizontal displacement associated with twinning shear (the shear results in vertical and horizontal displacements). Thus, shear off was not observed on the surface of the twinned pillar. As a result, in the nanoscale sample, the twin growth can be accomplished via the formation and migration of PB/BP interfaces, while the misfit strain in PB and BP interfaces can be released by emission of basal $\langle a \rangle$ dislocations. Furthermore, the image force due to free surfaces drives basal $\langle a \rangle$ dislocations out of the nanoscale sample.

In the wedged shaped pillar, we also refer to “geometry confinement” for creating a gradient stress field, which helps to form and capture twin embryos. To form a twin nucleus but stunt its subsequent propagation, a high stress needs to be localized within a small region. The high stress in a small region must be sufficient to form the twin, while the low stress in the surrounding region must be insufficient to expand, or grow, the twin. Thus, a truncated wedge-shaped geometry was designed to confine twin nucleation in the pillar top region. The narrower the pillar top, the smaller the twin embryo that can be captured.

The analysis on geometry confinement has been discussed in the Introduction and in the Section “Stress differences enabled isolation and in-situ visualization of twin nucleation and growth”.

4. The simulation images in fig.4 need to be enlarged in order that the embryo boundary structure can be discerned by the reader. Is this interface coherent?

Response: We have added the enlarged images of the twin boundaries in the Fig.4 (See the below images as well). The twin nucleation is distinct from twin growth. When the size of the twin embryo is below a critical value, the twin nucleates and is associated with forming coherent BP/PB interfaces. After a certain size, the twin can grow via semi-coherent BP/PB interfaces and coherent twin boundaries (R. C. Pond, et, all. Materials Research Letters, 4:4, 185-190). In Fig. 4a-f, the twin boundary is coherent. When the twin grows, its boundary in Fig. 4h-j is mostly coherent, but some regions are semi-coherent with misfit dislocations. The misfit dislocations are associated with the nucleation and emission of lattice dislocations with Burgers vector $\langle a \rangle$ on PB/BP interfaces (Fig. 4h-j). The nucleated dislocation glide on the basal plane into the matrix and the residual is left on the PB/BP interfaces (acting as a misfit dislocation). Thus, semi-coherent BP/PB interfaces formed partially in the twin boundaries.

The revised figures are provided below:

Figure 4. Molecular Dynamic simulations of twin nucleation. (a) Initial relaxed single crystal. (b-d) Nucleation of two twin nuclei at the corners on top surface. (e-h) Merge of two twin nuclei into a larger twin nucleus. (i-j) Propagation and growth of twin nucleus via migration of basal-prismatic (BP) and prismatic-basal (PB) interfaces. The symbol “T” indicates a basal $\langle a \rangle$ dislocation that is emitted from the PB interface. The inset images of (b-f) are magnified images in the pillar top region. The twin nucleus is surrounded by PB and BP interfaces. **The rectangles in each image indicate locations associated with the enlarged images.** In the enlarged images, the yellow dashed lines indicate the BP interfaces, and the green dashed lines indicate the PB interfaces. Atoms are color coded according to their excess energy, with the lowest energy atoms appearing blue, while the highest energy atoms appear red.

5. Have both the matrix and embryo been strained equally in fig. 5b? What is the meaning of CTB in such a bicrystal?

Response: Yes, in Fig. 5b, both matrix and embryo have been equally strained to form coherent lattice and complex structures. In such a bicrystal, the CTB deviates 1.9 degrees from the zero-stress CTB. This is related to the question #7, for which further details are provided.

On the page 20 (Fig. 5 caption), we have added the following statements: “Both matrix and embryo have been equally strained to form coherent lattice and complex structures.”; “The CTB deviates 1.9 degrees from the zero-stress CTB.”.

6. There is no analysis of the topological character of PB/BP “steps”: do they have dislocation character? Can they move laterally without diffusion? What defects are required to accommodate misfit? Would the presence of misfit dislocations impede migration of PB/BP boundaries?

Response: We appreciate this series of questions, which we address below, one at a time.

“do they have dislocation character?”

The PB/BP “steps” have dislocation character. PB/BP “steps” also contain the misfit on the PB/BP interface. A more precise description is that the PB/BP steps can be called coherency disclination (Gong, Wang, et. all, Materials Research Letters 5, 449-464 (2017); Hirth JP et al PNAS 117:196-204 (2020)).

“Can they move laterally without diffusion?”

The PB/BP “steps” are defined when the defect is viewed along the $\langle a \rangle$ direction. For lateral propagation of the “step”, people need to view the “steps” along the twinning shear direction (perpendicular to the $\langle a \rangle$ direction). However, the two ends of the “step” have not yet been characterized, although recent TEM characterization of twin boundaries in three dimensional

views revealed some facets or steps (Wang et al. Characteristic boundaries associated with three-dimensional twins in hexagonal metals, *Sci. Adv.* 2020; 6: eaaz2600). Our understanding is that diffusion is not necessarily associated with the lateral propagation of PB/BP “steps”. Further investigation is required but is beyond the scope of the current paper.

“What defects are required to accommodate misfit?”

The nucleation and emission of lattice dislocation with Burgers vector $\langle a \rangle$ on PB/BP interfaces likely accommodate the misfit and form semi-coherent PB/BP interfaces (Gong, Wang, et. all, *Materials Research Letters* 5, 449-464 (2017); Hirth JP et al *PNAS* 117:196-204 (2020)).

“Would the presence of misfit dislocations impede migration of PB/BP boundaries?”

Yes, as the twin nucleus expands, the misfit dislocations nucleated on the PB and BP boundaries can release the coherency stress and also, at the same time, further pin the PB and BP interfaces (see Fig. S13d-e).

The topological character of PB/BP ‘steps’ has been well summarized in the references (Pond, et. al, *Materials Research Letters*, 4:4, 185-190; Gong, Wang, et. all, *Materials Research Letters* 5, 449-464 (2017); Hirth, Wang, et. all, *Progress in Materials Science* 83 (2016) 417–471; Hirth JP et al *PNAS* 117:196-204 (2020)). We have modified the manuscript to include the topological analysis and have cited these references accordingly.

On page 23-24 in the revision: “In addition, we performed MD simulation and topological analysis for initial growth of a twin nucleus surrounded by PB and BP boundaries (see **Fig. S13a** and **Movie S6**). The topological character of PB and BP facets has been well summarized in the

references^{6, 30, 36, 37}. The PB and BP steps/facets have dislocation character and the misfit on the interface, have features of a coherency disclination^{6, 37}. Analogous to the Frank analysis for crystal growth⁶, we observe in calculation that the PB and BP facets grow faster (see **Fig. S13b-c**) than the $\{10\bar{1}2\}$ CTB during twin nucleation. As the twin nucleus expands, the misfit dislocations that must nucleate on the PB and BP facets to release the coherency stress also, at the same time, further pin the PB and BP boundaries (see **Fig. S13d-e**). This impedes migration of PB and BP facets and leaves the coherent twin bounded by slower growing $\{10\bar{1}2\}$ twin planes. As a result, the growing twin forms a lenticular shape, **Fig. S13f**, bounded by CTB terraces. This shape is consistent with the mature twin commonly found in bulk samples.”

7. Throughout the text, the orientation relationship between the matrix and embryo is 90° about $\langle 1-210 \rangle$. How does the additional small rotation about this axis arise in order that a true (mature) twin is formed?

Response: We appreciate the reviewer for pointing out this important aspect to improve the manuscript. We have added content in the revised manuscript on how a 90° BP/PB twin rotates to become a mature 86° twin, as noted below.

On page 23 in the revision: “First, there is an orientation difference between an embryo with the CPB/CBP boundary (90° between twin and matrix) and an “adult” $\{10\bar{1}2\}$ twin (86.22° between twin and matrix). An additional small rotation of the embryo into the twin orientation is needed to form a mature twin^{11, 33}. A CTB of large extent can be envisioned as a tilt wall of intrinsic dislocations and the rotation of 86.22° associated with the tilt wall is equally partitioned between the twin and matrix for a twin boundary of large extent^{11, 33}. However, in the case of a small nucleus, the stiffness of the surrounding matrix prevents this partitioning and all rotation (3.78°

derived from 90°) is imposed on the nucleus and is accommodated at the nucleus interface. The boundary of the rotated nucleus can be regarded as a disclination quadrupole with the disclination fields cancelling to first order. Hence, at the early stages of nucleation and growth, the rotation is not partitioned, and the interface is (0001) in the matrix parallel to $(1\bar{1}00)$ in the twin. That is, it is the true twin interface with the partitioning removed. The true twin boundary of large extent would be (0001) plus 1.9° in the matrix or $(1\bar{1}00)$ minus 1.9° in the twin, since the partitioned 1.9° is removed to the twin. If this removal of partitioning were not to occur in the large twin boundary, then large local elastic strains would be needed to make the interfaces the true (incoherent) twin interfaces. Thus, twin nuclei are bounded by CPB and CBP interfaces with attendant large coherency stresses partitioned to the nucleus because of the stiffness of the matrix.”

Reviewer #2 (Remarks to the Author):

This paper presents a breakthrough in visualization and validation of a twinning mechanism in magnesium, which has been proposed but unverified for the past 50 years. The most innovative design for experimental validation is the science-guided truncated wedge-shaped pillars (TWPs) from single-crystal Mg, which controls a steep stress field in the crystal under compression that can isolate twin nucleation from twin growth. With the TWPs, atomistic simulations and topological analysis were applied for a systematic study on the mechanism. The paper is well organized and professionally written. The microstructure characterization is excellent.

In the section of “Importance of the work”, the authors made it clear that the advance in understanding twinning mechanisms, especially early stage growth mechanisms, is of engineering

significance to develop novel methods in controlling/tuning twinning for materials. For micro/nano components, this can be straightforward. For macroscopic parts, it might not be easy to implement the geometry design strategy for stress control at nanoscale. It seems that the innovative method developed in this work does provide a direct way to study the alloying effect on twin nucleation/initiation in Mg, thus facilitating the discovery of high performance Mg alloys for engineering significance.

The methodology in this work can be widely used for other twinning mechanisms in any other solid materials. Moreover, it can have a broad impact in study of nanoscale deformation mechanisms. It is a paper worthy of publication as a highlight in Nature Communications.

Response: We highly appreciate the reviewer for recognizing the significance and contribution of this work, as well as the methodology we have developed. Twin formation in HCP materials has long been a controversial topic, but remains important to the materials community.

We have used this opportunity to carefully go through the entire manuscript. Changes are highlighted in yellow in the revised manuscript.

Again, we thank the reviewer for his or her efforts, as well as for the positive and helpful comments.

REVIEWERS' COMMENTS

Reviewer #1 (Remarks to the Author):

The authors have made helpful amendments to their manuscript which clarify the issues raised in my previous report. Publication of this fine work is recommended.